# PD-L1 Expression Associated with Epstein—Barr Virus Status and Patients’ Survival in a Large Cohort of Gastric Cancer Patients in Northern Brazil

**DOI:** 10.3390/cancers13133107

**Published:** 2021-06-22

**Authors:** Caroline de Fátima Aquino Moreira-Nunes, Cláudia Nazaré de Souza Almeida Titan Martins, Danielle Feio, Isamu Komatsu Lima, Leticia Martins Lamarão, Carolina Rosal Teixeira de Souza, Igor Brasil Costa, Jersey Heitor da Silva Maués, Paulo Cardoso Soares, Paulo Pimentel de Assumpção, Rommel Mário Rodríguez Burbano

**Affiliations:** 1Laboratory of Molecular Biology, Department of Clinical Medicine, Ophir Loyola Hospital, Belém, 66063-240 PA, Brazil; rhc@ophirloyola.pa.gov.br (C.N.d.S.A.T.M.); danielle.feio@ebserh.gov.br (D.F.); isamukomatsu@hotmail.com (I.K.L.); paulo.soares@uepa.br (P.C.S.); 2Laboratory of Pharmacogenetics, Department of Medicine, Drug Research and Development Center (NPDM), Federal University of Ceará, Fortaleza, 60430-275 CE, Brazil; 3Foundation Center for Hemotherapy and Hematology of Pará (HEMOPA), Department of Sorology, Belém, 66033-000 PA, Brazil; leticia.lamarao@hemopa.pa.gov.br; 4Human Cytogenetics Laboratory, Biological Science Institute, Federal University of Pará, Belém, 66075-110 PA, Brazil; carolrosalts@gmail.com; 5Department of Virology, Evandro Chagas Institute, Ananindeua, 67030-000 PA, Brazil; igorcosta@iec.gov.br; 6Hematology and Transfusion Medicine Center, Laboratory of Molecular and Cell Biology, Department of Medicine, University of Campinas, Campinas, 13083-970 SP, Brazil; jerseyhm@ufpa.br; 7Oncology Research Center, Department of Biological Sciences, Federal University of Pará, Belém, 66073-005 PA, Brazil; paulouf@ufpa.br

**Keywords:** gastric cancer, PD-L1, Epstein–Barr virus, biomarker

## Abstract

**Simple Summary:**

The programmed death ligand 1 (PD-L1) tumor expression is a manifestation of immune evasion and is used as a biomarker for treatment by immunotherapy and appears as a promising therapeutical option for Gastric Cancer (GC). The aim of this study was to evaluate the correlation between EBV status, PD-L1 expression and overall survival in a cohort of a thousand GC patients. Of the thousand tumors, 190 were EBV-positive and presented a high relative expression of PD-L1. the PD-L1 low expression was a characteristic of GC patients EBV-negative. We also show that the high expression of PD-L1 impacts on the survival probability and increase overall survival in GC patients. No EBV-negative GC patients had expression of PD-L1 protein, which suggests that the low expression is related to low overall survival. Patients with GC positive for EBV, presenting PD-L1 overexpression can benefit from immunotherapy treatments and performing the quantification of PD-L1 in gastric neoplasms should be adopted as routine.

**Abstract:**

Gastric cancer (GC) is a worldwide health problem, making it one of the most common types of cancer, in fifth place of all tumor types, and the third highest cause of cancer deaths in the world. There is a subgroup of GC that consists of tumors infected with the Epstein–Barr virus (EBV) and is characterized mainly by the overexpression of programmed cell death protein-ligand-1 (PD-L1). In the present study, we present histopathological and survival data of a thousand GC patients, associated with EBV status and PD-L1 expression. Of the thousand tumors analyzed, 190 were EBV-positive and the vast majority (86.8%) had a high relative expression of mRNA and PD-L1 protein (*p* < 0.0001) in relation to non-neoplastic control. On the other hand, in EBV-negative samples, the majority had a low PD-L1 expression of RNA and protein (*p* < 0.0001). In the Kaplan–Meier analysis, the probability of survival and increased overall survival of EBV-positive GC patients was impacted by the PD-L1 overexpression (*p* < 0.0001 and *p* = 0.004, respectively). However, the PD-L1 low expression was correlated with low overall survival in those patients. Patients with GC positive for EBV, presenting PD-L1 overexpression can benefit from immunotherapy treatments and performing the quantification of PD-L1 in gastric neoplasms should be adopted as routine.

## 1. Introduction

Gastric cancer (GC) is a major health problem, making it one of the most common types of cancer, in fifth place of all tumor types, and the third highest cause of cancer deaths in the world, affecting men at an average rate of two times more than in women [1,2]. The therapeutic context is negative, since the world average cumulative survival rate, five years after the diagnosis of the GC, is estimated at approximately 21% [3,4].

For northern Brazil, without considering non-melanoma skin tumors, GC is the second and fifth most frequent type of cancer in men and women, respectively [5]. GC was subclassified, by the Cancer Genome Atlas Research Network, due to molecular differences between cancer cells into four main subtypes [6]. The main subgroup represents the tumors that harbor chromosomal instability (CIN); in this subgroup is located about 50% of all GC patients. In the second subgroup are tumors that present microsatellite instability-high (MSI-H), and about 22% of patients present these molecular patterns. In the third subgroup, representing about 20% of all tumor cases, tumors are classified as genomically stable. The fourth and last subgroup represents all tumors that present the Epstein–Barr virus (EBV) infection [7] and those characterized mainly by the overexpression of the programmed cell death protein-ligand-1/2 (PD-L1/2) [8].

Is widely known that tumor cells are able to express a large number of co-inhibitory immune ligands that promote an immune evasion in the tumor microenvironment that leads to progression and metastasis. Programmed death ligand 1 and 2 (PD-L1 and PD-L2) are the two ligands to the programmed cell death protein-1 (PD-1) receptor. PD-L1 is largely expressed in immune and tumor cells, while antigen-presenting cells are responsible for expressing the PD-L2 ligand. In the tumor microenvironment, a large amount of tumor-infiltrating lymphocytes (TILS) play an important role in the release of IFN-γ, and thus, induce the expression activation of PD-L1 in the tumor cells and in stromal and blood cells [9,10]. In the tumor microenvironment, tumor cells express PD-L1 and bind to the PD-1 receptors that are presented in the activated T cells that reach the tumor, and this interaction generates a suppression immune signal, disabling T cells of destroying the cancer cells, and affecting the humoral immune responses to the tumor [11].

In this scenario, the presence of PD-L1 expression in EBV-positive GC tumors can benefit those patients with immunotherapy, once testing the expression of PD-L1 is the current standard in most solid tumors: several studies have evaluated clinical results according to the status of expression of PD-L1 in GC [12]. The aim of this study was to evaluate the correlation between EBV status, histopathological data, PD-L1 relative quantification expression, and overall survival in a large cohort of GC patients in northern Brazil.

## 2. Methods

### 2.1. Patients and Sample Collection

Between 2005 and 2015, a total of 1000 GC patients who underwent surgical resection were included in this cohort. The removed tumor samples were carefully dissected, and the tumor tissues and corresponding non-neoplastic tissues of the gastric mucosa were collected to perform a comparative study of the tumor tissue with its non-tumor correspondent, which was used as a control. That is, there were 2000 paired samples (1000 tumor and 1000 controls). The pathological staging of the GC followed the classification of tumors, nodules, metastasis (TNM) of the 8th American Joint Cancer Committee/Union for International Cancer Control (AJCC/UICC) [13].

In previous evaluations before surgery and for tumor staging, all the patients included in this study underwent chest X-ray, abdominal ultrasound, or an abdominal computed tomography (CT) scan. Gastrectomy was performed based on the distance between the cardia and the tumor. For distal GC, the subtotal gastrectomy is the standard procedure, while for proximal GC total gastrectomy is the common procedure. Regarding the extension of lymphadenectomy, at least one dissection of gastrectomy D1 (dissection of all nodules in Group 1) was performed for early GC, while at least one dissection D2 (defined as dissection of all nodules in Group 1 and Group 2) was applied in cases of advanced GC. For D2 dissections, a combined organ resection was sometimes performed to obtain a curative resection. All patients’ procedures followed were in accordance with Helsinki guidelines and regulations. This study was approved by the Ethics Committee of Ophir Loyola Hospital (no. 2,798,615).

### 2.2. Patient’s Follow-Up

For the patient’s follow-up and survival analysis, the assessments were performed every 3 months, after 3 years of surgery and after that period, the assessments were performed for every 6 months until the patient’s death. The follow-up procedures included the review of the following parameters: physical examinations, medical records, common blood tests, chest radiography, abdominal ultrasound, and computed tomography. In case of tumor recurrence, those were diagnosed mainly by biopsies or imaging, when biopsies were not possible to obtain. Patients with tumor recurrence were eligible to receive chemotherapy based on 5-fluorouracil (FU). All patients included in this study did not receive any previous preoperative chemotherapy.

### 2.3. Sample Processing

For molecular and expression analysis, biological material was conserved in liquid nitrogen shortly after surgical procedure. The DNA was extracted with phenol-chloroform, after the tumor tissue underwent a maceration procedure [14]. For pathological analysis, the tumor samples were fixed in paraformaldehyde, before being included in a slide with paraffin and stained with hematoxylin and eosin.

### 2.4. Detection of H. pylori and cagA

To detect the presence of *H. pylori* in all patient’s gastric samples a commercially rapid urease test (Promedical, Juiz de Fora, Minas Gerais, Brazil) was performed according to kit instructions. The following procedures were applied before urea hydrolysis: samples were placed in a tube containing 20 g/L of Christensen’s urea agar and incubated at 37 °C for 24 h. The Polymerase Chain Reaction (PCR) in agarose electrophoresis gel was performed to confirm all negative results and to evaluate the presence of the cagA gene in *H. pylori* positive samples, all samples were tested in duplicate. The oligonucleotides used here have been described by Covacci and Rappuoli [15].

### 2.5. EBV Sample Detection

EBV presence in gastric tumor samples was detected using a biotinylated primer probe (5′-AGACACCGTCCTCACCACCCGGGACTTGTA-3′), which is complementary to the small EBV-encoded RNA-1 (Eber1) [16]. For EBV detection, the RNA in situ hybridization (ISH) technique was performed, and for signal amplification we used an anti-biotin mouse antibody (DakoCytomation^®^, Carpinteria, CA, USA) and a rabbit anti-biotinylated immunoglobulin (DakoCytomation^®^, Carpinteria, CA, USA).

The streptavidin-biotin peroxidase complex (DakoCytomation^®^, Carpinteria, CA, USA) and the chromogen diaminobenzidine (DakoCytomation^®^, Carpinteria, CA, USA) were used to detect this reaction. The slides were contrasted with Harris’ hematoxylin and the cells were examined under light microscopy at 40× magnification. In this examination, 10 representative microscopic fields containing at least five cells were evaluated. The epithelial cells were considered positive for EBV presence when 5% or more cells were stained in brown/red. Both tumor and non-neoplastic samples from the same patient were analyzed.

### 2.6. mRNA Relative Quantification

The total mRNA of the 1000 GC patients was isolated from tumor tissues and their paired adjacent non-tumor tissues (confirmed by pathology as non-tumor). RNA was obtained using Tri-reagent^®^ (Thermo Fisher Scientific, Carlsbad, CA, USA), according to the manufacturer’s protocol. The RNA conversion into cDNA was performed using the High Capacity^®^ cDNA Reverse transcription kit (Thermo Fisher Scientific, Carlsbad, CA, USA), following the manufacturer’s instructions. The cDNA was amplified by quantitative real-time PCR (RT-qPCR) using TaqMan probes (Thermo Fisher Scientific, Carlsbad, CA, USA) and amplified on a 7500 Real-Time machine (Applied Biosystems, Carlsbad, CA, USA). All RT-qPCR sample reactions were analyzed in triplicate. The cycle threshold (Ct) and expression values with standard deviations were calculated. The GAPDH gene was selected as an internal control for RNA entry and reverse transcription efficiency. All RT-qPCRs were performed in triplicate for the target gene (PD-L1 Hs01125297_m1) and the internal control (GAPDH: NM_002046.3).

The gene expression relative quantification was calculated according to the guidelines [17]. The control sample of non-neoplastic gastric tissue, adjacent to the tumor analyzed, was used as a calibrator for each tumor. When the relative quantification was 1.00, it meant that the amount of PD-L1 mRNA is the same in the control sample and in the tumor. When the relative quantification was 1.50 it meant that the amount of PD-L1 mRNA is 50% greater in the tumor compared to the control sample. For this reason, all tumor samples with a value equal to or greater than 1.50 were considered highly expressed, and tumor samples whose relative quantification was less than 1.5 were considered to have low PD-L1 expression.

### 2.7. Protein Expression Analysis

The total proteins of the paired samples (tumor tissues and non-neoplastic tissues) were extracted using the M-PER regent for the extraction of mammalian proteins (Thermo Fisher Scientific, Carlsbad, CA, USA). The purified proteins of samples were separated by SDS-PAGE and electrotransferred on a PVDF membrane (Hybond-P, GE Healthcare, USA) for detection and expression. The PVDF membrane was blocked with phosphate-buffered saline containing 0.1% Tween 20 and 5% low-fat milk and incubated overnight at 4 °C with the corresponding primary antibodies: anti-PD-L1 (clone MIH1, Thermo Fisher Scientific, Carlsbad, CA, USA) and anti-beta actin (Thermo Fisher Scientific, Carlsbad, CA, USA).

After several washing steps with TBS, a secondary antibody conjugated to peroxidase was added for 1 h at room temperature. The immunoreactive bands were visualized using the Western blotting reagent Luminol, and the images were acquired using an ImageQuant 350 digital imaging system (GE Healthcare, Danderyd, Sweden). Beta-actin (ACTB) was used as a loading control and the relative quantification of protein expression was calculated following the same basis as the quantification of mRNA, that is, a control sample of non-neoplastic gastric tissue adjacent to the analyzed tumor, was designated as a calibrator for each tumor.

### 2.8. Statistical Analysis

Pearson’s chi-square χ^2^ test and Fisher’s exact test were used to assess continuous variables. The data are shown as the median and interquartile range (IQR) depending on the normality of the samples, the Shapiro–Wilk test was applied to determine whether the samples followed a normal distribution. For comparison analysis between two or more different groups, corresponding nonparametric tests were applied. Linear regression was calculated with the ggpubr R package and were considered significant with *p* < 0.05. In the quantification of mRNA and PD-L1 protein, they were only considered significant when these two biomarkers simultaneously had *p*-values < 0.05.

The overall survival of patients was calculated from the date of diagnosis to the date of death or the last follow-up assignment, using the Kaplan–Meier method with the log-rank statistical. All graphs and statistical analysis were done by using R (https://www.r-project.org, accessed on 20 March 2021).

## 3. Results

GC samples were collected from 1000 patients between 2005 and 2015. The average age was 58 years (range, 20–82 years). The number of young patients, under 64 years of age, was statistically higher (61.8%) than patients older than 64 years of age (*p* < 0.0001) and the number of female patients (34.2%) was statistically lower (*p* < 0.0001) than the number of male patients (65.8%). The number of patients presenting with the initial disease, in stage I (14.3%), was statistically lower (*p* < 0.0001) than all stages, mainly stage IV, the most advanced (38.3%).

Lauren’s classification [18] also shows a significant difference (*p* < 0.0001) between the diffuse (41.2%) and intestinal (58.8%) subtypes. Regarding tumor location, tumors located in cardia were significantly more frequent (*p* < 0.0001) than others in different locations in the stomach. There was no statistical difference between the number of patients with and without metastases in other organs. Table 1 summarizes the clinicopathological data of patients included in this study.

Regarding infections by *H. pylori* and EBV, we can observe that the presence of the bacterium was significant (*p* < 0.0001): 82.8% of the patients presented with infection by *H. pylori*, and 61.9% (*p* < 0.0001) of the patients had the virulence conferred by the presence in the genome of the bacterium of the cagA gene. In the case of EBV infection, the absence of the virus in most of the studied samples is significant (*p* < 0.0001), only 19.0% of the studied samples had the viral genome (Table 1).

Overall, 810 samples were EBV-negative and 190 EBV-positive. Of the 190 EBV-positive samples, 165 tumors (86.8%) had high relative PD-L1 mRNA expression, and 25 tumors had low PD-L1 mRNA expression (13.2%) (*p* < 0.0001) in relation to the non-neoplastic control. Regarding protein expression, 149 (78.4%) had high expression and 41 (21.6%) low protein expression (*p* < 0.0001) In the EBV-negative samples, the vast majority had RNA expression and low PD-L1 protein (*p* < 0.0001) (Figure 1, Table 2). There was a significantly greater number of GC patients with diffuse-type and EBV-positive status (poorly differentiated) than EBV-positive in GC patients intestinal-type (*p* < 0.0001) (Table 2).

In Figure 2A, we can observe that the presence of the EBV virus in the tumor sample is associated with a high relative expression of mRNA and PD-L1 protein, in relation to the non-neoplastic control (above 1.5 or 50%). The absence of the EBV virus (Figure 2B) in the tumor sample is strongly associated with a low expression of PD-L1 mRNA and protein. Regarding histological types with EBV-positive status, the expression profiles of PD-L1 mRNA and protein in patients with diffuse-type GC are significantly higher (*p* < 0.0001) than the expression profiles of the intestinal type. In the analysis of patients with EBV-negative status, those of the diffuse and intestinal-type do not have a statistically significant expression of mRNA and protein (*p* > 0.05) (Figure 2A,B).

There is a linear correlation of the increase in mRNA and PD-L1 protein levels in EBV-positive patients with a diffuse histological type (R = 0.90) and intestinal-type (R = 0.86) (Figure 2C,D, respectively), showing samples of greater overexpression of PD-L1 in diffuse type. On the other hand, the linear correlation of increased levels of PD-L1 mRNA and protein in EBV-negative patients is low in both the diffuse (R = 0.09) and intestinal (R = 0.19) histological types (Figure 2E,F, respectively), showing a totally different distribution than EBV-positive samples.

Table 2 shows that the presence of EBV was significant in patients under 64 years (*p* < 0.0001), with diffuse histological type tumors located in the cardia (*p* < 0.0001) and infected with *H. pylori* cagA positive (*p* < 0.05). EBV-negative patients often have metastases in other organs and low expression and production of PD-L1 (Figure 3A). In comparison, EBV-positive patients do not have metastases, and EBV-positive expression and production is high (Figure 3B). EBV patient’s negative cells often have lymph node metastases (N1–N3) and have low expression and production of PD-L1 (Figure 3C), on the other hand, EBV-positive patients frequently did not have lymph node metastasis (N0) and have high expression and production of PD-L1 (*p* < 0.001) (Figure 3D). The correlations of presence/absence of EBV and the presence/absence of lymph node metastases can be better visualized by distribution ranking (in percentage) in the donut chart graphs in Figure 3E,F.

The high expression of PD-L1, regardless of EBV infection, was significantly higher (*p* < 0.05) in stage I/II tumors, with no lymph node metastases (N0) and in other organs (M0) (*p* < 0.001). Overexpression was significant in tumors of the diffuse subtype (*p* < 0.05), located in cardia (*p* < 0.001). There was no statistical association with the presence or absence of *H. pylori* cagA positive bacteria. These results are summarized in Table 3.

In Table 4, we can see that EBV-negative patients <64 years have less production of PD-L1 than patients >64, this phenomenon is reversed in EBV-positive patients, where younger patients have a higher production of PD-L1. Samples from EBV-negative male patients produce more PD-L1 protein than female patients. In EBV-positive patients, there is no statistical difference between the production of PD-L1 between genders. This difference is also not found in EBV-negative patients between the production of PD-L1 and tumor staging, however, the association of EBV-positive patients with high production of PD-L1 and tumor staging I/II is significant.

The presence/absence of lymph node and other organ metastases in EBV-negative patients is not associated with the production of PD-L1, whereas EBV-positive samples have a higher production of PD-L1 in samples from patients without lymph node metastasis or in other organs. Regarding the histological type, EBV-negative patients do not show differences in the production of PD-L1, always remaining below 1.5 or 50%. On the other hand, samples of diffuse EBV-positive tumors produce greater expression of PD-L1 (*p* < 0.001) than intestinal tumors, although both histological types are over-expressed for PD-L1 when infected with this virus.

Table 4 also shows that the levels of PD-L1 mRNA and protein stratified by samples obtained from different stomach locations are significantly higher (*p* < 0.001) in EBV-positive patients (Figure 4A) compared to EBV-negative patients (Figure 4B). An interesting fact is that, regardless of EBV infection, tumors located in a specific region of the stomach, such as cardia and gastric fundus, have a higher expression than tumors located in multiple sectors of the (*p* < 0.0001). The levels of mRNA and protein of PD-L1 stratified in samples obtained from different locations of the stomach are significantly higher in EBV-negative patients whose tumor is located in the gastric pylorus (*p* < 0.001) in relation to the other locations (Figure 4B), although as commented earlier, as EBV-negative, the expression of mRNA and protein of these tumors is below 1.5 (50%).

Regarding the increased production of PD-L1 and EBV status, only samples from EBV-negative patients were associated with low production of PD-L1 protein and the presence of positive *H. pylori* cagA, however, this association is not repeated in PD-L1 mRNA expression, and for this reason, it will not be considered. EBV-positive patients were not associated with the amount of mRNA and protein produced by the tumor cell and the presence of virulent *H. pylori* cagA positive strains (Table 4).

The overall survival (in months) of GC patients with diffuse and intestinal histological types were analyzed by Kaplan–Meier curve. In Figure 5A,B, we can see that the high expression of PD-L1 in patients with diffuse and intestinal histological types, impacts on the survival probability and increase overall survival (Log-rank, *p* < 0.0001, *p* = 0.004, respectively). However, the low expression of PD-L1 accentuates the low overall survival in patients with diffuse GC more than in the intestinal histological type.

The expression data are described in Table 2; the PD-L1 low expression is a characteristic of the EBV-negative GC, as described in this study. No EBV-negative patients had expression of PD-L1 protein (0.0%), which suggests that the low expression is related to low overall survival.

In relation to patients with GC in stage I/II, we can see in Figure 5C that EBV-positive patients have a longer survival than the same EBV-negative patients (Log-rank, *p* = 0.0039). In Figure 5D, the phenomenon is repeated, but the difference is more pronounced, with EBV-positive patients having a longer survival than the same EBV-negative patients with stage III/IV, (Log-rank, *p* < 0.0001). When comparing stages I/II with III/IV (Figure 5C vs. Figure 5D) as expected, early-stage patients survive longer, but an interesting fact is that EBV-positive patients in stages III and IV have practically the same survival as Stage I and II EBV-negative patients. The log-rank test with *p* < 0.05 was used to differentiate the groups.

The comparison of the overall survival (in months) between patients with diffuse and intestinal histological GC performed by Kaplan–Meier analysis are shown in Figure 6. Only the patients with EBV-positive status and with the diffuse type presented a longer survival than the intestinal subtype (*p* = 0.0032).

## 4. Discussion

EBV can directly cause the development of GC, and the prevalence of EBV associated with GC varies from 4 to 18%, with a frequency twice as high in males [19]. The incidence of EBV-positive GC varies between 1 to 30% in different world regions, with an average rate of 10% worldwide. In our study, the prevalence was 19.0%, also twice as often in males (66.8%) compared to females (33.2%) [20,21,22]. In this present study we found a similar incidence (20%), previously reported in another study from our group, performed in the same population and in a smaller sample (*n* = 302) [23], this incidence was similar to other studies in the United States and Germany (16–26%) [24,25,26]. In other countries, however, lower frequencies were found [27,28,29], thus, showing that the EBV prevalence can vary widely across geographic regions.

EBV shows an anatomical preference during gastric tumorigenesis and is predominantly present in the proximal stomach, with reported rates of 11.6% in the cardia and 9.5% in the stomach body [22,30]. A Korean study by Park et al. [31] showed that 84.4% of the positive GC were in the upper or middle third of the stomach. In the present study, this phenomenon was repeated, where 44.7% of the EBV-positive GCs were in cardia, while in the gastric body housed 24.7% of the tumors. In the upper or middle third of the stomach, 76.2% of malignancies associated with EBV were detected (Table 1).

GC associated with EBV is associated with a positive prognosis, however, this association is less common in advanced or metastatic conditions [6]. In our study, we can also observe the phenomenon that EBV-positive patients without lymph node metastasis had a significantly higher proportion than patients with lymph node metastasis (*p* < 0.001). Regarding metastases in other organs, we can observe a significantly lower number of cases associated with EBV with metastases than EBV-positive patients with no metastases (Table 2).

In addition, the overall survival analysis (at 60 months) of GC patients clearly shows a higher survival of EBV-positive patients when compared to EBV-negative patients. Although the number of EBV-negative cases was found to be significantly higher using Fisher’s exact test (*p* = 0.056), in stage IV cases, the latter finding can be explained because the public hospital where the samples were obtained serves patients belonging to a more needy population in the region, which normally seeks hospital services in advanced stages of cancer, and therefore, a greater number of advanced cases is always expected.

In particular, EBV-positive tumors often exhibit overexpression of PD-L1 [32]. This overexpression was also found in gastric tumors associated with EBV (*p* < 0.001), in this study (Table 3 and Table 4, Figure 1, Figure 2A–C and Figure 3B,D). Many research groups have described that the level of PD-L1 expression ranges from 34 to 92% approximately in GCs associated with EBV [33,34,35,36]. The PD-L1 expression and its association with gastric cancer prognosis is a controversial topic. In this study, overexpression of PD-L1 was significantly associated with a good prognosis since we found increased production of PD-L1 in samples from patients without lymph node metastases or in other organs. Additionally, as can be seen in Table 2, the low expression of PD-L1 is characteristic of the EBV-negative GC, including that no EBV-negative patient had a high expression of PD-L1 protein (0.0%), which indirectly reveals that the low expression of PD-L1 is related to low overall survival.

A recently published review revealed that most of the studies (a total of 15) described the PD-L1 expression as a negative prognostic factor for overall survival, and only three studies associated it as a positive prognostic factor, with one study reporting no prognostic significance [37]. However, recently, overexpression of PD-L1 was significantly associated with a good prognosis, where Kim et al. [38] described that PD-L1 protein levels increased patient survival in a positive correlation, while another study described PD-L1 expression was also associated with progression-free survival in patients with metastatic GC [39]. Additionally, overall survival was higher in patients with higher PD-L1 tumor expression (positive combined score ≥ 1) than tumors with lower PD-L1expression (positive combined score < 1 tumors) [40].

Our findings are also corroborated by the study by Moore et al. [41], who found that the subtype of GC EBV-positive is indicative of a good prognosis and is more prevalent in young patients, as in the present study, where the number of young patients is statistically higher than patients older than 64 years of age (*p* < 0.0001). Another study with a large number of samples (*n* = 3241) also found results similar to our findings, EBV-positive patients (5%) were young and male, with GC samples histologically poorly differentiated, and patients having a good prognosis when expressing the PD-L1 protein [38]. The difference is in the number of EBV-positive patients, since in our study it was approximately four times higher (19%), but as previously discussed, although the rate was high, it was still within the expected distribution [19,22].

PD-L1 protein expression is used as a prognostic biomarker for treatment by immunotherapy. Immunotherapy appears as a promising novel treatment for patients with metastatic or recurrent GC. The PD-L1 tumor expression is described as a manifestation of immune evasion. As mentioned, the distinctive characteristic of GC positive for EBV, in relation to EBV-negative, is the overexpression of PD-L1. The positivity of PD-L1 in GC positive for EBV can be effectively directed towards immunotherapy; testing the expression of PD-L1 should therefore become routine in GC [12,42].

Immunotherapy based on PD-L1 inhibitors are an important innovation in tumor therapy in recent years, which can reactivate T cell immune functions, increase the body’s immune response capacity and, ultimately, allow the immune system to eliminate gastric cancer tumor cells effectively and achieve tumor remission. Inhibition of the programmed death-1 (PD-1)/programmed-ligand 1 (PD-L1) axis with immune control point inhibitors (ICPI), including nivolumab, avelumab and pembrolizumab, has emerged as new therapeutical strategy for GC patients. Treatment with ICPI in GC provides a survival benefit and anti-PD-1 treatment can especially improve the overall survival rate and prolong the duration of the response. In addition, anti-PD-1/PD-L1 therapy is likely to be more effective in subgroups with high microsatellite instability, EBV-positive, or high mutation load in advanced GC [12,43].

The targeted therapy scenario for GC is promising, and many clinical trials evaluating immune response to those therapies combined with targeted molecular agents are generating a great deal of enthusiasm. In addition, genomic data from The Cancer Center Genome (TCGA) and Asian Cancer Research Group (ACRG) classifications are being widely explored to identify molecular features of tumor subtypes to allow, in the near future, clinical trials that might include GC patient populations enriched with distinct biomarkers [44].

GC molecular classification has emerged as an important tool to select the right patients who will benefit from those targeted therapies or benefits of an accurate disease prognosis. The addition of the new molecular classification strategy to a current classic subtype can be a promising option, particularly EBV virus stratification. The findings of predictive biomarkers, such as PD-L1, should improve patient diagnosis and therapy in the era of personalized medicine, once targeted therapies can potentially change the GC patients’ treatment scenario, leading to a better understanding of heterogeneity and better results.

## 5. Conclusions

Patients with GC positive for EBV and presenting PD-L1 overexpression can benefit from immunotherapy treatments and performing the quantification of PD-L1 in gastric neoplasms should be adopted as routine.

## Figures and Tables

**Figure 1 cancers-13-03107-f001:**
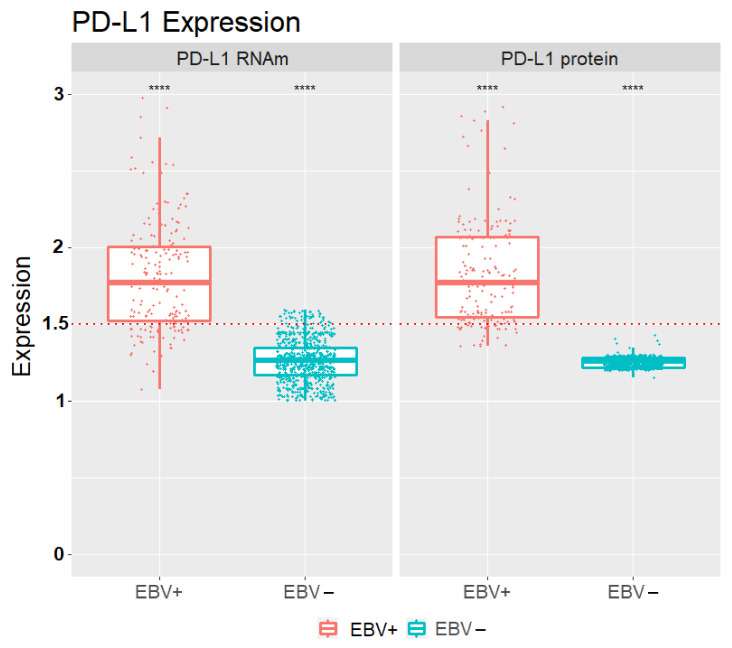
PD-L1 mRNA level expression in patients with positive and EBV-negative status. (**** *p* < 0.0001). The red dotted lines represent the 1.5-fold change.

**Figure 2 cancers-13-03107-f002:**
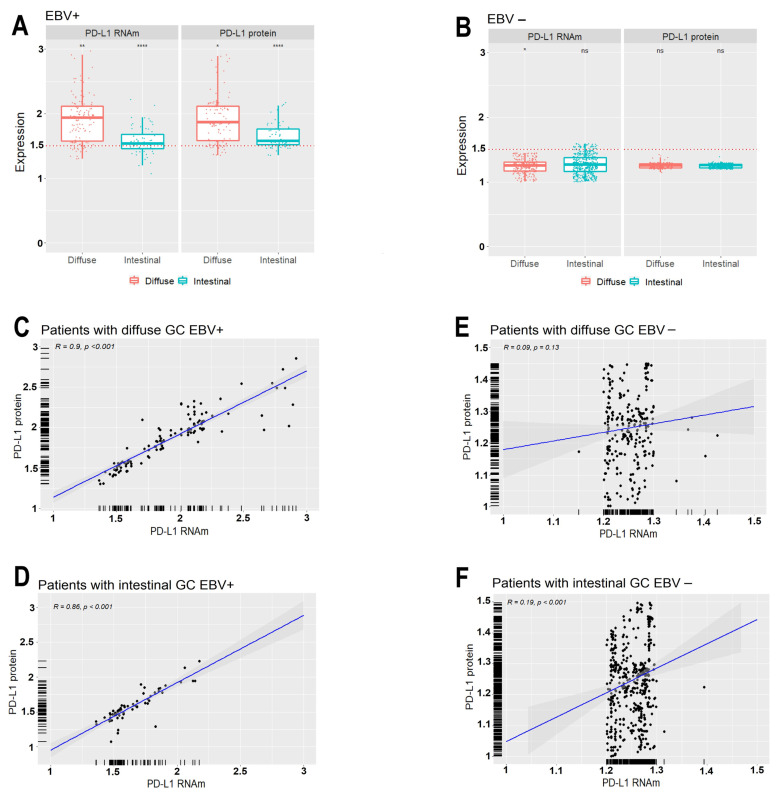
(**A**,**B**) Relative normalized expression of PD-L1 mRNA and protein in tumor gastric tissue of patients with diffuse and intestinal gastric cancer with positive and EBV-negative status presented in Box plots. The red dotted lines represent the 1.5-fold-change. (ns, not significant, * *p* < 0.05, ** *p* < 0.01, **** *p* < 0.0001). The horizontal lines inside the boxes represents the median of expression values. The vertical lines above and below the box delineate the maximum and minimum values, and the dots indicate outliers. (**C**,**D**) Linear correlation of increased PD-L1 mRNA and protein levels in EBV-positive patients with diffuse histology (R = 0.90, *p* < 0.01) and intestinal (R = 0.86, *p* < 0.01). (**E**,**F**) Linear correlation of increased PD-L1 mRNA and protein levels in EBV-negative patients with diffuse histological (R = 0.09, *p* = 0.13) and intestinal (R = 0.19, *p* < 0.01).

**Figure 3 cancers-13-03107-f003:**
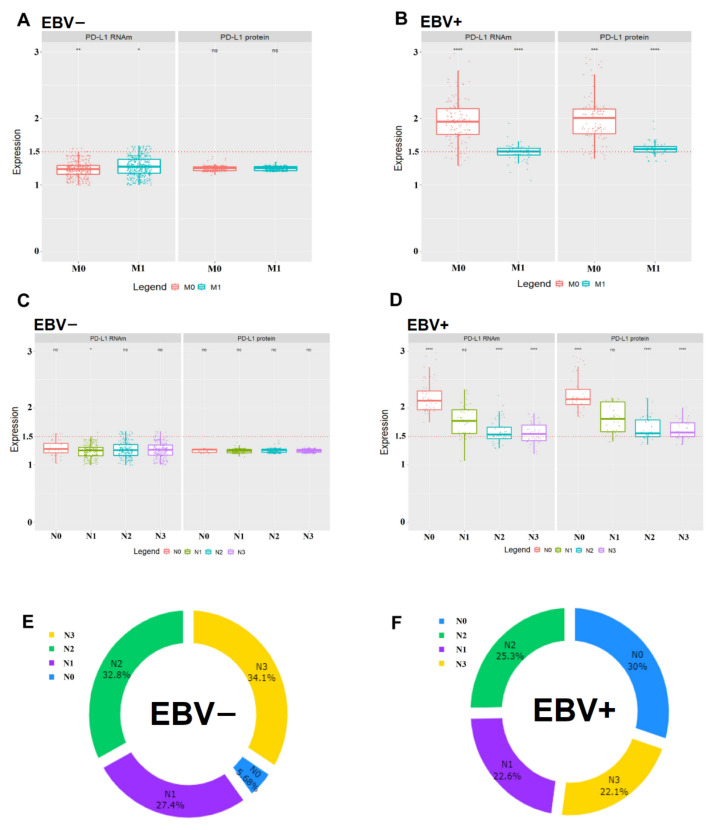
(**A**,**B**) Normalized relative expression of PD-L1 mRNA and protein in the patients’ gastric tumor tissue without metastasis (M0) and with metastasis (M1) (*p* < 0.001) stratified in EBV− and EBV+ status. The 1.5-fold-change is represented by the red dotted lines. (ns, not significant, * *p* < 0.05, ** *p* < 0.01, *** *p* < 0.001, **** *p* < 0.0001) (**C**,**D**) PD-L1 mRNA and protein normalized relative expression in the gastric tumor tissue of patients with lymph node metastasis N0, N1, N2, and N3 (*p* < 0.001). The horizontal line inside the box represents median values. Maximum and minimum values are represented by vertical lines above and below the box delineate, and the dots indicate outliers. (**E**,**F**) Donut Chart shows the ranking of distributions (%) in samples from EBV− and EBV+ patients with Lymph node metastasis N0, N1, N2, and N3.

**Figure 4 cancers-13-03107-f004:**
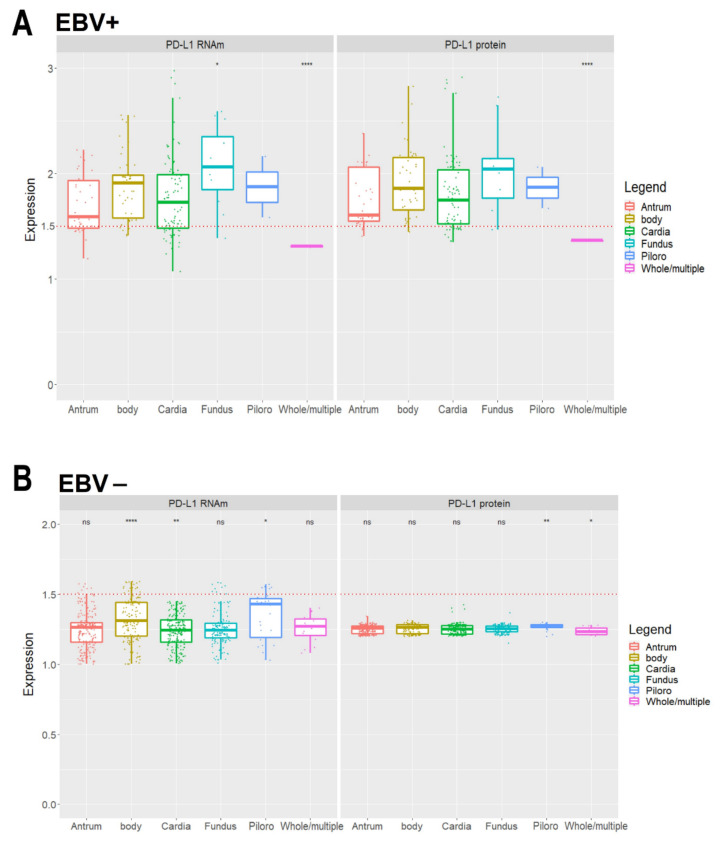
PD-L1 mRNA and protein levels stratified in samples obtained from different stomach locations. Levels of PD-L1 mRNA and protein in patients with EBV-positive status (**A**,**B**) EBV-negative status (ns, not significant, * *p* < 0.05, ** *p* < 0.01, **** *p* < 0.0001). The red dotted lines represent the 1.5-fold-change.

**Figure 5 cancers-13-03107-f005:**
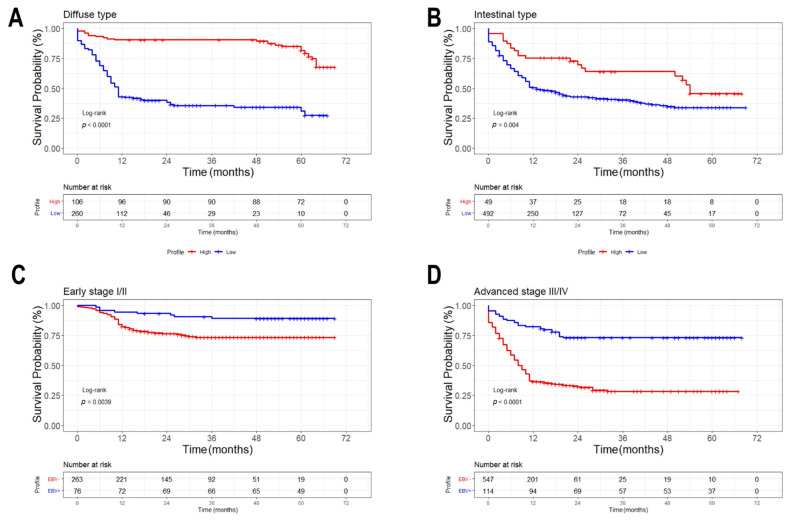
Overall survival (in months) by Kaplan–Meier analysis of the of gastric cancer patients with diffuse and intestinal histological type associated with tumor stage. (**A**) Gastric cancer patients with diffuse histological type that express PD-L1, and (**B**) Gastric cancer patients with intestinal histological type that express PD-L1. We described a high gene expression (≥1.5; red line), as opposed to low expression (<1.5; blue line) for the association of a lower probability of survival. (**C**) Patients with gastric cancer in stage I/II and (**D**) stage III/IV. The log-rank test with *p* < 0.05 was used to differentiate the groups.

**Figure 6 cancers-13-03107-f006:**
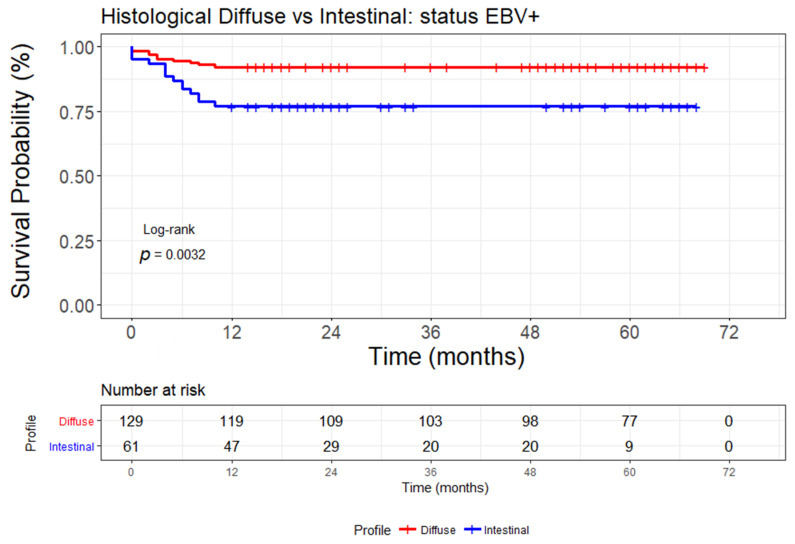
Overall survival (in months) analysis by Kaplan–Meier curves of gastric cancer patients with diffuse and intestinal histological types only in patients with EBV-positive (EBV+) status. Patients with diffuse GC have longer survival than patients with intestinal GC. The log-rank test with *p* < 0.05 was used to differentiate the groups.

**Table 1 cancers-13-03107-t001:** Clinicopathological Features.

Variable	No of Patients (%)	*p*-Value (χ^2^ Test)
Age		
<64 years	618 (61.8)	<0.0001
≥64 years	382 (38.2)	
Gender		
Male	658 (65.8)	<0.0001
Female	342 (34.2)	
Stage		
I	143 (14.3)	<0.0001
II	196 (19.6)	
III	278 (27.8)	
IV	383 (38.3)	
Presence of Lymph node metastasis		
N_0_	103 (10.3)	<0.0001
N_1_	265 (26.5)	
N_2_	314 (31.4)	
N_3_	318 (31.8)	
Metastasis		
Absent	487 (48.7)	0.411
Present	513 (51.3)	
Histological type *		
Diffuse	412 (41.2)	<0.0001
Intestinal	588 (58.8)	
Tumor location		
Antrum	238 (28.3)	<0.0001
Body	202 (20.2)	
Cardia	312 (31.2)	
Fundus	156 (15.6)	
Pylorus	28 (2.8)	
Whole/multiple	19 (1.9)	
*H. Pylori* Urease/PCR		
Negative	172 (17.2)	<0.0001
Positive	828 (82.8)	
CagA		
Negative	381 (38.1)	<0.0001
Positive	619 (61.9)	
EBV		
Negative	810 (81.0)	<0.0001
Positive	190 (19.0)	

* According to Lauren’s Classification [18].

**Table 2 cancers-13-03107-t002:** EBV status and clinicopathological features.

Clinical Features	Sample Size, (%)	EBV−	EBV+	*p*-Value (χ^2^ Test)	*p*-Value (Fisher Exact Test)
Patients	1000	810 (81.0%)	190 (19.0%)		
Age					
<64 years	618 (61.8)	478 (59.0)	140 (73.7)	<0.0001 ***	<0.0001 ***
≥64 years	382 (38.2)	332 (41.0)	50 (26.3)		
Gender					
Male	658 (65.8)	531 (65.6)	127 (66.8)	0.737	0.403
Female	342 (34.2)	279 (34.4)	63 (33.2)		
Stage					
I	143 (14.3)	108 (13.3)	35 (18.4)	0.112	0.056
II	196 (19.6)	155 (19.1)	41 (21.6)		
III	278 (27.8)	236 (29.1)	42 (22.1)		
IV	383 (38.3)	311 (38.4)	72 (37.9)		
Lymph node metastasis					
N_0_	103 (10.3)	46 (5.7)	57 (30.0)		
N_1_	265 (26.5)	222 (27.4)	43 (22.6)	0.001 **	0.001 **
N_2_	314 (31.4)	266 (32.8)	48 (25.3)		
N_3_	318 (31.8)	276 (34.1)	42 (22.1)		
Metastasis					
Absent	487 (48.7)	360 (44.4)	127 (66.8)	0.001 **	0.001 **
Present	513 (51.3)	450 (55.6)	63 (33.2)		
Histological type					
Diffuse	412 (41.2)	283 (34.9)	129 (67.9)	<0.0001 ***	<0.0001 ***
Intestinal	588 (58.8)	527 (65.1)	61 (32.1)		
Tumor location					
Antrum	238 (28.3)	242 (29.9)	41 (21.6)	<0.0001 ***	<0.0001 ***
Body	202 (20.2)	155 (19.1)	47 (24.7)		
Cardia	312 (31.2)	227 (28.0)	85 (44.7)		
Fundus	156 (15.6)	143 (17.7)	13 (6.8)		
Pylorus	28 (2.8)	26 (3.2)	2 (1.1)		
Whole/multiple	19 (1.9)	17 (2.1)	2 (1.1)		
Urease/PCR					
Negative	172 (17.2)	108 (13.3)	64 (33.70	<0.0001 ***	<0.0001 ***
Positive	828 (82.8)	702 (86.7)	126 (66.3)		
CagA					
Negative	381 (38.1)	297 (36.7)	84 (44.2)	0.054	<0.05 *
Positive	619 (61.9)	513 (63.3)	106 (55.8)		
mRNA (PD-L1)					
High	165 (16.5)	45(5.6)	165 (86.8)	<0.0001 ***	<0.0001 ***
Low	835 (83.5)	765 (94.4)	25 (13.2)		
Protein (PD-L1)					
High	194 (19.4)	0.0 (0.0)	149 (78.4)	<0.0001 ***	<0.0001 ***
Low	806 (80.6)	810 (100)	41 (21.6)		

* *p*-value < 0.05, ** *p*-value < 0.01 or *** *p*-value < 0.001.

**Table 3 cancers-13-03107-t003:** Association of PD-L1 expression and clinicopathological features.

Clinical Features	Sample Size (%)	mRNA (PD-L1) Mean ± SD	*p*-Value	Protein (PD-L1) Mean ± SD	*p*-Value
All Patients’	1000				
Age					
<64 years	618 (61.8)	1.41 ± 0.38	0.266	1.37 ± 0.33	0.050
≥64 years	382 (38.2)	1.31 ± 0.16		1.36 ± 0.47	
Gender					
Male	658 (65.8)	1.38 ± 0.33	0.609	1.39 ± 0.45	<0.05
Female	342 (34.2)	1.36 ± 0.29		1.34 ± 0.25	
Stage					
I/II	339 (33.9)	1.44 ± 0.43	<0.05 *	1.42 ± 0.35	<0.05 *
III/IV	661 (66.1)	1.34 ± 0.25		1.34 ± 0.41	
Lymph node metastasis					
N_0_	103 (10.3)	1.90 ± 0.67	<0.05 *	1.79 ± 0.49	<0.05 *
N_1_	265 (26.5)	1.34 ± 0.23		1.36 ± 0.58	
N_2_	314 (31.4)	1.31 ± 0.16		1.32 ± 0.19	
N_3_	318 (31.8)	1.29 ± 0.13		1.30 ± 0.17	
Metastasis					
Absent	487 (48.7)	1.47 ± 0.45	<0.001	1.45 ± 0.37	<0.05 *
Present	513 (51.3)	1.28 ± 0.10		1.30 ± 0.15	
Histological type					
Diffuse	412 (41.2)	1.49 ± 0.45	<0.05 *	1.45 ± 0.37	<0.05 *
Intestinal	588 (58.8)	1.29 ± 0.13		1.31 ± 0.39	
Tumor location					
Antrum	238 (28.3)	1.32 ± 0.20	<0.001 **	1.31 ± 0.22	<0.001 **
Body	202 (20.2)	1.42 ± 0.37		1.45 ± 0.30	
Cardia	312 (31.2)	1.42 ± 0.38		1.41 ± 0.58	
Fundus	156 (15.6)	1.33 ± 0.33		1.32 ± 0.27	
Pylorus	28 (2.8)	1.31 ± 0.16		1.38 ± 0.23	
Whole/multiple	19 (1.9)	1.25 ± 0.04		1.26 ± 0.09	
Urease/PCR					
Negative	172 (17.2)	1.51 ± 0.46	<0.001 **	1.46 ± 0.39	<0.001 **
Positive	828 (82.8)	1.34 ± 0.28		1.35 ± 0.39	
CagA					
Negative	381 (38.1)	1.40 ± 0.36	0.276	1.37 ± 0.31	0.449
Positive	619 (61.9)	1.36 ± 0.30		1.37 ± 0.43	
EBV					
Negative	810 (81.0)	1.25 ± 0.03	<0.001 **	1.27 ± 0.33	<0.001 **
Positive	190 (19.0)	1.90 ± 0.45		1.80 ± 0.35	

* *p*-Value < 0.05 or ** *p*-Value < 0.01.

**Table 4 cancers-13-03107-t004:** Clinicopathological features and gene expression of PD-L1 in gastric cancer.

Clinical Features	EBV-Negative					EBV-Positive				
	Sample Size	PD-L1 mRNA RQ (Median ± IQR)	*p*-Value	PD-L1 Protein RQ (Median ± IQR)	*p*-Value	Sample Size	PD-L1 mRNA RQ (Median ± IQR)	*p*-Value	PD-L1 Protein RQ (Median ± IQR)	*p*-Value
Age										
<64 years	478 (59.0)	1.24 ± 0.30	<0.001	1.23 ± 0.11	<0.001	140 (73.7)	1.98 ± 0.49	<0.001	1.87 ± 0.36	<0.001
≥64 years	332 (41.0)	1.25 ± 0.03		1.33 ± 0.49		50 (26.3)	1.68 ± 0.19		1.60 ± 0.22	
Sex										
Male	531 (65.6)	1.25 ± 0.03	0.719	1.28 ± 0.40	0.021 *	127 (66.8)	1.92 ± 0.47	0.774	1.82 ± 0.38	0.785
Female	279 (34.4)	1.25 ± 0.03		1.24 ± 0.11		63 (33.2)	1.84 ± 0.41		1.77 ± 0.28	
Stage										
I/II	263 (32.4)	1.25 ± 0.03	0.799	1.27 ± 0.13	0.124	76 (40.0)	2.09 ± 0.51	<0.001	1.96 ± 0.37	<0.001
III/IV	547 (67.5)	1.25 ± 0.03		1.27 ± 0.39		114 (60.0)	1.78 ± 0.35		1.70 ± 0.30	
Lymph node metastasis										
N_0_	46 (5.7)	1.25 ± 0.03	0.172	1.29 ± 0.13	0.110	57 (30.0)	2.42 ± 0.44	<0.001	2.19 ± 0.27	<0.001
N_1_	222 (27.4)	1.24 ± 0.03		1.28 ± 0.59		43 (22.6)	1.81 ± 0.23		1.77 ± 0.26	
N_2_	266 (32.8)	1.25 ± 0.03		1.27 ± 0.14		48 (25.3)	1.63 ± 0.19		1.59 ± 0.19	
N_3_	276 (34.1)	1.25 ± 0.02		1.26 ± 0.13		42 (22.1)	1.61 ± 0.15		1.55 ± 0.17	
Metastasis										
Absent	360 (44.4)	1.26 ± 0.03	0.561	1.26 ± 0.47	<0.001	127 (66.8)	2.08 ± 0.45	<0.001	1.96 ± 0.32	<0.001
Present	450 (55.6)	1.25 ± 0.03		1.28 ± 0.14		63 (33.2)	1.54 ± 0.10		1.49 ± 0.12	
Histological type										
Diffuse	283 (34.9)	1.25 ± 0.03	0.608	1.24 ± 0.10	0.05 *	129 (67.9)	2.03 ± 0.49	<0.001	1.91 ± 0.36	<0.001
Intestinal	527 (65.1)	1.25 ± 0.09		1.28 ± 0.40		61 (32.1)	1.64 ± 0.19		1.58 ± 0.21	
Tumor location										
Antrum	242 (29.9)	1.25 ± 0.04	<0.001	1.24 ± 0.12	<0.001	41 (21.6)	1.75 ± 0.26	0.002 **	1.70 ± 0.27	0.002 **
Body	155 (19.1)	1.25 ± 0.03		1.32 ± 0.15		47 (24.7)	1.99 ± 0.43		1.87 ± 0.31	
Cardia	227 (28.0)	1.25 ± 0.03		1.27 ± 0.58		85 (44.7)	1.89 ± 0.48		1.78 ± 0.38	
Fundus	143 (17.7)	1.25 ± 0.04		1.24 ± 0.11		13 (6.8)	2.29 ± 0.59		2.07 ± 0.37	
Pylorus	26 (3.2)	1.26 ± 0.02		1.34 ± 0.17		2 (1.1)	1.87 ± 0.27		1.87 ± 0.40	
Whole/multiple	17 (2.1)	1.23 ± 0.02		1.25 ± 0.09		2 (1.1)	1.36 ± 0.00		1.31 ± 0.01	
PCR urease										
Negative	108 (13.3)	1.25 ± 0.03	0.186	1.23 ± 0.09	0.053	64 (33.70)	1.97 ± 0.49	0.345	1.85 ± 0.40	0.314
Positive	702 (86.7)	1.25 ± 0.03		1.27 ± 0.35		126 (66.3)	1.87 ± 0.43		1.78 ± 0.32	
CagA										
Negative	297 (36.7)	1.25 ± 0.03	0.718	1.24 ± 0.11	<0.05 *	84 (44.2)	1.93 ± 0.47	0.611	1.82 ± 0.37	0.481
Positive	513 (63.3)	1.25 ± 0.03		1.29 ± 0.40		106 (55.8)	1.88 ± 0.44		1.78 ± 0.33	

*p*-value by Kruskal–Wallis test. * *p* < 0.05 or ** *p* < 0.001, significantly different between groups. *n*: number of samples; RQ, relative quantification, in which the matched non-neoplastic sample was designated as a calibrator from each neoplastic sample; T/*n*, the ratio of protein expression between tumor samples and matched non-tumor tissue; IQR, interquartile range; CagA gene, cytotoxicity-associated gene A; EBV, Epstein–Barr virus. The patients’ survival time was determined by the time interval between the date of surgery and the date of death.

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
