# Peer review of "PD-L1 Expression Associated with Epstein—Barr Virus Status and Patients’ Survival in a Large Cohort of Gastric Cancer Patients in Northern Brazil"

_cancers, 2021, doi:10.3390/cancers13133107_

Round 1
Reviewer 1 Report
In this paper, the authors have analyzed a thousand gastric tumors for the expression of PD-L1 from patients of Brazil. PD-L1 expression was analyzed in dissected tumor cells and compared to adjacent non-tumor cells. The presence of EBV was also analyzed in the tumor cells with à probe for EBER. They examined these two parameters in function of different features of patients including age and various clinical manifestations. The major observation is that high PD-L1 expression correlated with a better probability of survival and the presence of EBV in tumors cells. This is an interesting study, well-conducted and discussed. Presumably, the high expression of PD-L1 reflects that the immune response is at work (and may explained the better outcome). Thus, immunotherapy treatments blocking the PD1-PD-L1 pathway could be beneficial for these patients as discussed by the authors.
Specific comments:
1) It seems that the authors sometimes used PD1 in place of PD-L1. This is confusing like in the introduction lines 51-67. This should be modified.
2) Abstract lines 26-28, it seems that the two sentences are repetitive or maybe the authors referred to patients with EBV+ tumors in the second one, but this is not indicated. This should be clarified.
3) lines 211-213. The sentence is difficult to understand. It should be rephrased and clarified.
Author Response
Dear reviewer, my co-authors and I would like to thank you for the suggestions made during this high-quality review and then we present the answers to the questions.
We inform that with the reviews and suggestions, we were able to improve the idea presented by our work and we appreciate the opportunity. We hope this review has left the article suitable for publication in this high-impact journal and respect in the area.
Kind Regards.
Response to reviewer 1
1) It seems that the authors sometimes used PD1 in place of PD-L1. This is confusing like in the introduction lines 51-67. This should be modified.
R = We are sorry about that misunderstanding. We rephrase all this section to be clearer: “Is widely known that tumor cells are able to express a large number of co-inhibitory immune ligands that promote an immune evasion in the tumor microenvironment that leads to progression and metastasis. Programmed death ligand 1 and 2 (PD-L1 and PD-L2) are the two ligands to programmed cell death protein-1 (PD-1) receptor. PD-L1 ligand is largely expressed in immune and tumor cells, while antigen presenting cells are responsible for express the PD-L2 ligand. In the tumor microenvironment, are a large amount of tumor infiltrating lymphocytes (TILS) that play an important role in release IFN-γ and thus, induce the expression activation of PD-L1 in the tumor cells, and in stromal and blood cells [9, 10]. In the tumor microenvironment, tumor cells express PD-L1 and binds to the PD-1 receptor that are presented in the activated T cells that reach the tumor, and this interaction generates a suppression immune signal, disabling T cells of destroying the cancer cells, affecting the humoral immune responses to tumor”
2) Abstract lines 26-28, it seems that the two sentences are repetitive or maybe the authors referred to patients with EBV+ tumors in the second one, but this is not indicated. This should be clarified.
R = we rephrase the sentence to be clearer: “In the Kaplan-Meier analysis, the probability of survival and increased overall survival of EBV-positive GC patients was impacted by the PD-L1 overexpression (p <0.0001 and p = 0.004, respectively). However, the PD-L1 low expression was correlated with low overall survival in those patients.”
3) lines 211-213. The sentence is difficult to understand. It should be rephrased and clarified.
R = we rephrase the sentence to be clearer: “There was a significantly greater number of GC patients with diffuse-type and EBV-positive status (poorly differentiated) than EBV-positive in GC patients intestinal-type (p <0.0001)”.

Reviewer 2 Report
I have read with great pleasure and interest this original article.
the topic was very relevant, the study was well designed and conducted.
language was clear and easy to understand (only minor check required).
results were clearly described and presented as well as interesting.
I have no concerns about the publication of this paper in its present form (just minor language revisions).
Author Response
Response to reviewer 2
Dear reviewer, my co-authors and I would like to thank you for the suggestions made during this high-quality review. We did a careful review and adjusted the small writing mistakes made in the article. We hope this review has left the article suitable for publication in this high-impact journal and respect in the area.
Kind Regards.
